# MoVE identifies metabolic valves to switch between phenotypic states

Naveen Venayak[1], Axel von Kamp[2], Steffen Klamt[2] & Radhakrishnan Mahadevan [1,3]

Metabolism is highly regulated, allowing for robust and complex behavior. This behavior can often be achieved by controlling a small number of important metabolic reactions, or metabolic valves. Here, we present a method to identify the location of such valves: the metabolic valve enumerator (MoVE). MoVE uses a metabolic model to identify genetic intervention strategies which decouple two desired phenotypes. We apply this method to identify valves which can decouple growth and production to systematically improve the rate and yield of biochemical production processes. We apply this algorithm to the production of diverse compounds and obtained solutions for over 70% of our targets, identifying a small number of highly represented valves to achieve near maximal growth and production. MoVE offers a systematic approach to identify metabolic valves using metabolic models, providing insight into the architecture of metabolic networks and accelerating the widespread implementation of dynamic flux redirection in diverse systems.

---

[1] Department of Chemical Engineering and Applied Chemistry, University of Toronto, 200 College Street, Toronto, ON M5S 3E5, Canada. [2] Max Planck Institute for Dynamics of Complex Technical Systems, Sandtorstraße 1, 39106 Magdeburg, Germany. [3] Institute of Biomaterials and Biomedical Engineering, University of Toronto, 164, College Street, Toronto, ON M5S 3G9, Canada. Correspondence and requests for materials should be addressed to R.M. (email: krishna.mahadevan@utoronto.ca)

Biological regulatory networks allow for metabolic transitions which are apparent in a range of biological processes[1,2]. These regulatory systems are the basis for ubiquitous cellular phenomena including complex cell cycles, robustness to changing environments, and eukaryotic development through stem cell differentiation and tissue morphogenesis. The architecture of these systems can vary tremendously, from regulating short pathways, to having global metabolic effects[3,4]. Engineered applications of these systems have been developed to understand complex cellular processes[5], implement genetic logic[6,7], and improve microbes to produce valuable chemicals[8–13], using control points identified by biochemical assay, or intuited from models of metabolic structure. The choice of such control points is further complicated for non-natural objectives, such as chemical production, where few biological examples exist.

Metabolic network structure has been studied using metabolic models for diverse purposes including consolidating high-throughput -omics data, identifying drug targets, and predicting metabolic phenotypes[14]. In particular, these models have been used extensively for designing microbial cell factories[15]. This is commonly accomplished using mixed-integer linear programming (MILP) techniques to identify network modifications, which can then be implemented by modulating gene function. In particular, a large number of algorithms have been developed to identify such interventions to improve microbes for growth-coupled chemical production[15–18]. These strategies rely on the simultaneous production of both biomass and product; however, the burden imposed by high-flux production pathways can severely limit this strategy when producing chemicals at high yields, which is particularly evident when considering the trade-offs between yield and productivity[9,12,19]. Furthermore, growth-coupling can require specific network features, which may not exist in all organisms[20–22]. Instead, growth and production could be separated, allowing production processes to be operated in two stages, where biomass is accumulated before high-rate production is initiated. A phenotypic shift can be realized using a number of stimuli including inducers[23,24], internal metabolites[25,26], and cell density[27,28]. Although many strain design algorithms exist, these algorithms identify static interventions to achieve a single steady-state growth-coupled phenotype. None of these algorithms is suitable for identifying dynamic interventions, where multiple phenotypes must be considered. In this paper, we present a novel and systematic approach to identify metabolic valves and apply it for the production of 87 metabolites that can be produced by the genome-scale model of *Escherichia coli*.

## Results

### Overview of metabolic valve enumerator (MoVE). The efficient transition between phenotypes could be achieved by controlling flux through a set of metabolic valves, effectively decoupling both phenotypes. By designating a target reaction for each phenotype, MoVE uses a constraint-based metabolic model to identify (1) a set of static knockouts and (2) a set of dynamically controlled valves, to enable the transition between high flux for each of these targets (Fig. 1a). Static knockouts can be implemented using genome editing, and valves controlled using responsive genetic elements[6,7] (Fig. 1b) to enable, for example, the transition between growth and production states. These knockouts minimally impact the first desired target, but prime the network for the activity of valves. This set of valves can then be used to eliminate undesired fluxes, and enforce a high production yield (Fig. 1c). For some products, the phenotype can be shifted using process conditions such as pH, temperature, or oxygen availability, which trigger native regulatory systems[29,30] (Fig. 1d); however, these systems may not coincide with engineering

objectives such as chemical production. Instead, internal metabolites or inducers can trigger sensors and metabolic controllers to manipulate valves and effect a phenotypic shift (Fig. 1e). Since these strategies enforce a predefined minimum product yield, adaptive evolution can be effectively applied (Fig. 2).

**Core model strategies in *E. coli*.** We first apply MoVE to identify strategies to decouple growth from chemical production using a core reconstruction of *E. coli*[31]. We present a strategy for the production of α-ketoglutarate (AKG), an important intermediate in the tricarboxylic acid (TCA) cycle with uses as a dietary supplement, to illustrate the role of valves and knockouts for redirecting flux, and their impact on the phenotypic space (Fig. 3). This strategy achieves theoretical maximum production of AKG in the production state. Such core model strategies have recently been applied successfully for the production of itaconic acid using an iterative approach[32]. A more efficient two-stage strategy can be directly identified by MoVE, and this strategy has been shown to be effective for improving the yield and titer of itaconic acid, in addition to overcoming the need for media supplementation due to auxotrophy[13]. This production strategies illustrates the tight competition between product and biomass precursors, common to many target molecules.

**Genome-scale strategies in *E. coli*.** Next, we applied this algorithm to a genome-scale metabolic reconstruction of *E. coli*, iJO1366[33]. Despite the high computational demand of many strain design algorithms applied to genome-scale models, recent algorithmic advancements[18], modern MILP solvers, and high-performance computing clusters can be used to explore a large range of target metabolites. We use a distributed MILP algorithm to identify intervention strategies which meet our desired growth and production thresholds, identifying the optimal feasible solution within a fixed computational time for each metabolite (Supplementary Fig. 1).

We searched for intervention strategies for all 87 organic products that can be derived from glucose in our model (Supplementary Table 1). In this case, we focus on natural chemicals; however, this analysis can be trivially extended for the case of non-natural chemicals by including heterologous reactions and exchanges in the model. We investigate strategies for two scenarios: full and partial decoupling. We have proposed that an optimal operating strategy will generally require full decoupling, with a switch from maximum growth to maximum production[9]; however, substrate uptake rate can decrease in resting cells[19] and the inability to produce biomass precursors could lead to difficulties for sustained production. For these reasons, it may be beneficial to sacrifice product yield to maintain some capacity for cell growth, leading to partially decoupled production. For these simulations, fully decoupled strategies achieve over 90% of theoretical maximum product yield at the expense of cell growth, while partially decoupled strategies achieve over 70% of theoretical maximum yield, while maintaining a minimum biomass yield of 0.01 gdw/mmol (approximately 10% of the maximum biomass yield, allowing a growth rate of $0.1\,h^{-1}$). Both strategies achieve over 90% of theoretical maximum biomass yield in the growth state. Alternatively, an intermediary stage could be included for high expression of the production pathway at the end of the growth stage (e.g., using an inducible promoter), to reduce the need for heterologous protein expression in the production stage.

First, we explore the ability of single valves to redirect flux for full decoupling, obtaining over 90% biomass yield and product yield in their respective states (Fig. 4a). We identified strategies where controlling single valves could meet the desired flux

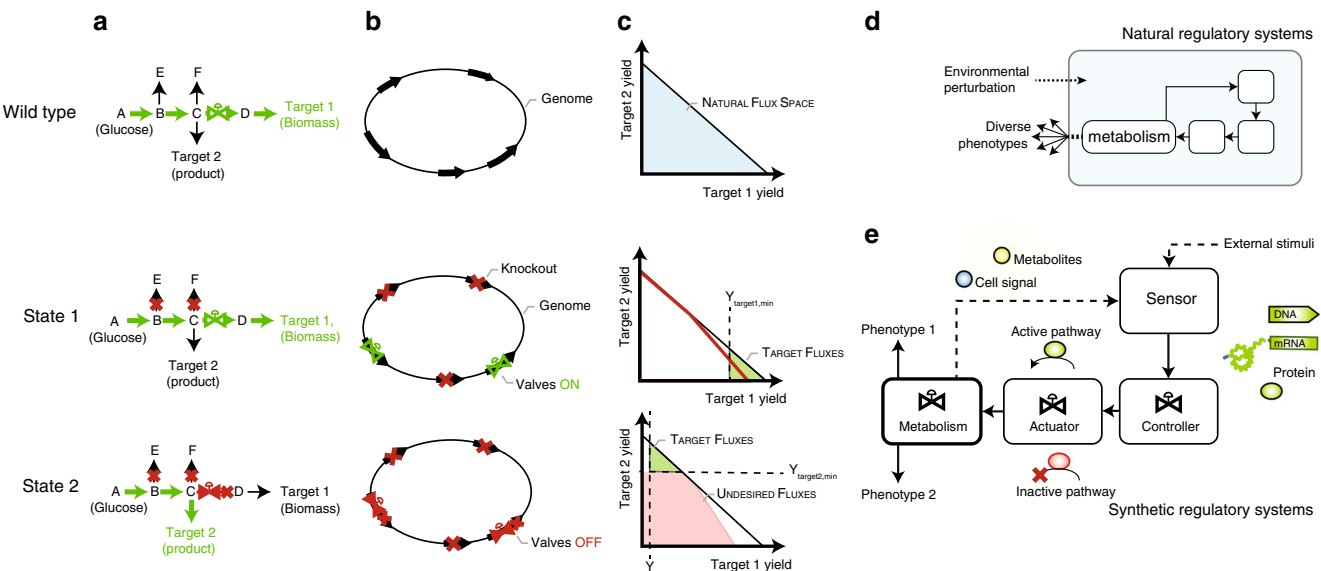

**Fig. 1** Overview of the metabolic valve enumerator. **a** MoVE identifies intervention strategies to decouple growth and production phenotypes using a set of static genetic knockouts and dynamically regulated metabolic valves. Knockouts prime the network to allow usage of these valves to redirect metabolism between states. **b** These conceptual valves can be implemented through genetic engineering, deleting genes providing undesired network functionality and dynamically regulating a set of valves. **c** The steady-state stoichiometric flux space is mapped on two-dimensions to illustrate the trade-off between metabolic targets. The wild-type has a wide range of potential phenotypes, often achieving maximum biomass yield using natural regulatory systems. In the first state, flux is maintained through valves to achieve a high yield of the first target. In the second state, flux through valves and undesired flux vectors are eliminated (red), enforcing a desired phenotype (green). **d** Native regulatory systems are able to sense some environmental perturbations to exhibit controlled phenotypic responses. **e** Synthetic control logic can be implemented using a typical control architecture and applying genetic circuits to sense metabolites and control flux through targeted metabolic valves toward specified objectives

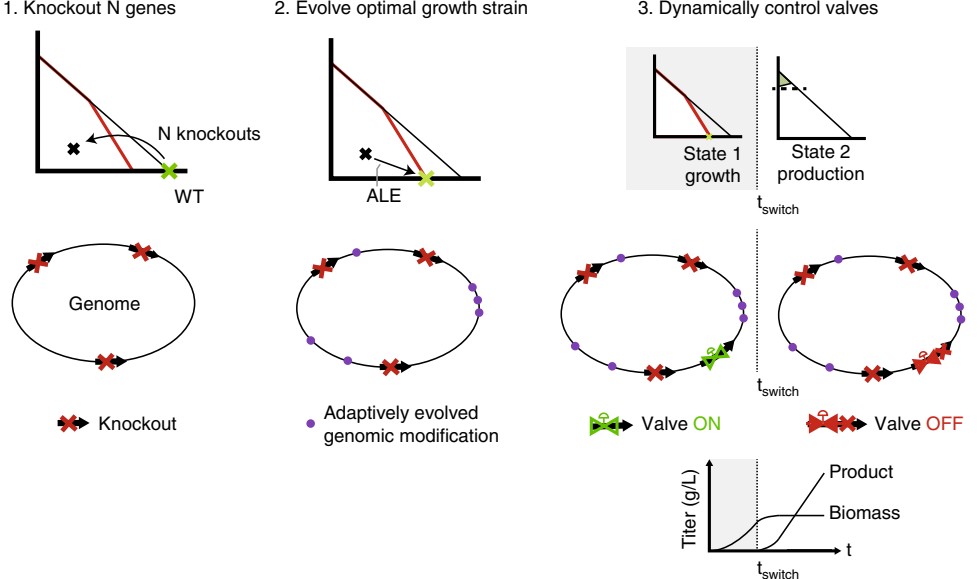

**Fig. 2** Implementation considerations for MoVE strategies. First, a set of static knockouts can be made using canonical genetic engineering techniques. Second, if the growth rate has suffered due to sub-optimality, the strain can be adaptively evolved to maximize growth rate, accumulating genomic modifications. Finally, genetic circuitry can be applied to effect a switch by eliminating flux through target valves, and achieving at least the minimum specific target (e.g., product yield)

thresholds for 56 products, or 64% of all targets (Fig. 4b, Supplementary Figs. 3 and 4). Three metabolic subsystems were highly represented: glycolysis, the TCA cycle, and oxidative phosphorylation. The top five valves included three from glycolysis: glyceraldehyde-3-phosphate dehydrogenase (GAPD, *gapA*), pyruvate dehydrogenase (PDH, *aceE*) and phosphogluco-mutase (PGM, *pgm*); citrate synthase (CS, *gltA*) from the TCA cycle; and oxygen exchange (EX_o2(e), *passive*) used in oxidative phosphorylation. These valves were relatively evenly distributed amongst the degree of reaction connectivity, indicating their high representation is not solely owed to their branched nature (Supplementary Fig. 2). However, we noted a high representation of reactions proximal to the 12 precursor metabolites related to the naturally evolved bow-tie (hourglass) topology of

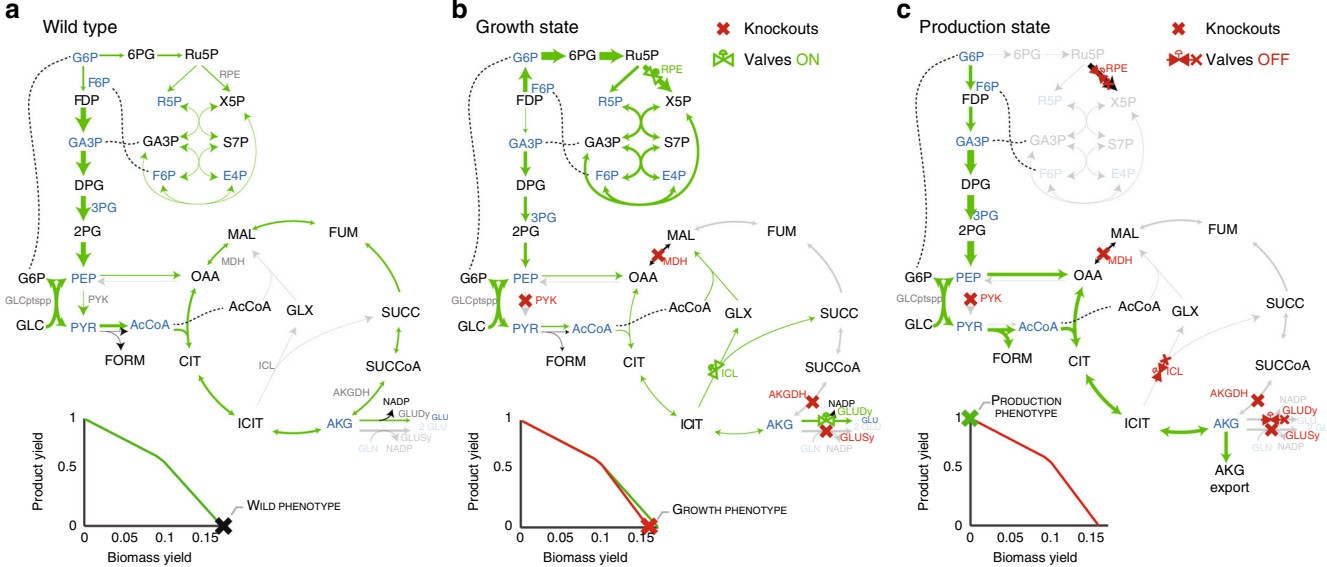

**Fig. 3** Application of MoVE to a core model for α-ketoglutarate (AKG) production. Line thickness is proportional to pathway flux, metabolites highlighted in blue are biomass precursors. **a** Wild-type flux distribution of the *E. coli* core model. Aerobically grown wild-type *E. coli* is known to partition flux between glycolysis and the pentose phosphate pathway, with a fully active tricarboxylic acid (TCA) cycle and respiratory chain[56], leading to maximal biomass yield and growth rate. **b** Flux-distribution with identified knockouts applied. Following knockout of the predicted reactions and associated genes: pyruvate kinase (PYK, *pykAF*), succinyl-CoA synthetase (SUCOAS, *sucA*), glutamate synthase (GLUSy, *gltB*), and malate dehydrogenase (MDH, *mdh*), flux is partitioned toward the pentose phosphate pathway from glycolysis, and flux through the TCA cycle is redirected through the glyoxylate shunt, marginally impacting the maximum biomass yield. **c** Flux-distribution after eliminating flux through knockouts and valves, achieving maximal AKG yield

metabolism[34,35], indicating this topology may be important to allow for efficient metabolic transitions.

Next, we identify strategies for partial decoupling. These strategies maintain a growth rate of $0.1\,\mathrm{h}^{-1}$ in the production state, and target a more modest 70% of maximum product yield (Fig. 4c). Experimentally determined essential reactions[36] were also blacklisted from being used as valves, corresponding to the goal of allowing a minimum biomass yield throughout. We identified valves from similar metabolic subsystems for both full and partial decoupling, with a few exceptions (Fig. 4d, Supplementary Figs. 5 and 6). First, PGM had a notably higher representation for partial decoupling, likely due to the essentiality of many other glycolytic reactions. In addition, more valves from upper glycolysis and the pentose phosphate pathway are identified. Lastly, α-ketoglutarate dehydrogenase (AKGDH, *sucA*) is the only single valve identified in the TCA cycle, and connected to AKG, a precursor to amino acids (Fig. 4e). The requirement to maintain a minimum biomass yield is a strong constraint, requiring the production of several metabolites as biomass precursors; thus, more complex strategies are required to ensure the low production of these metabolites while ensuring high product yield.

With the optimal valves for each product identified, we interrogated whether the top five valves from both decoupling strategies could be used for a broader range of products (Fig. 5). Although some of these valves could be effective for many products, this trend was not universal. For example, employing oxygen exchange as a valve could decouple fewer than 10 products, including known fermentative products such as acetate, ethanol, lactate, and succinate. This strategy has been applied for these products by exploiting the natural switch between high yield aerobic growth and low-yield fermentative growth (coupled to high-yield product synthesis), triggered by oxygen availability and controlled at the process-level. Interestingly, these strategies required relatively few knockouts, indicating that metabolic networks are structured to readily allow for such natural

transitions. However, these results indicate that employing oxygen availability as a valve may only be applicable for a small subset of relevant products, motivating the implementation of synthetic genetic circuits to control metabolic flux. Similarly, while GAPD was one of the most frequently identified valves, it could only decouple 11 products.

Contrarily, we have identified valves which could be applied to a majority of tested products. For example, CS was identified as an effective valve to decouple 55 products. It lies at an important branchpoint which has been successfully dynamically controlled to improve the production of isopropanol from acetyl-CoA[37]. It is an intuitive choice for eliminating the main pathway for flux into the TCA cycle, leading to overflow production of desired compounds. We have shown that this valve is applicable to more than 60% of tested metabolites, making it a good candidate for modular platform strains. This reaction is also known to be regulated by global regulators, with reduced flux during anaerobic growth to compensate for increased flux from pyruvate to fermentative products[38]. In addition, two closely related valves: PDH and PGM, were also both effective for a wide range of products. They are central to committing phosphoenolpyruvate or pyruvate to the TCA cycle. PDH is known to be downregulated in anaerobic conditions due to oxygen sensitivity of the *lpdA* subunit, and its activity replaced by pyruvate formate lyase (PFL, *pflAB*). These reactions are also important for controlling ATP generation through pyruvate kinase (PYK, *pykAF*) and NADH generation through PDH, allowing alternate routes for entry into the TCA cycle. These results highlight the importance of the pyruvate node for controlling metabolic flux.

AKGDH, which produces succinyl-CoA from AKG, was uniquely identified as a valve for partial decoupling. It is proximal to a highly regulated branch point for the production of amino acids from AKG, which also has implications in nitrogen metabolism and cofactor balance. We identified AKGDH as a suitable valve for AKG production, as well as glutamate, arginine, and proline which are derived from AKG. Another important valve

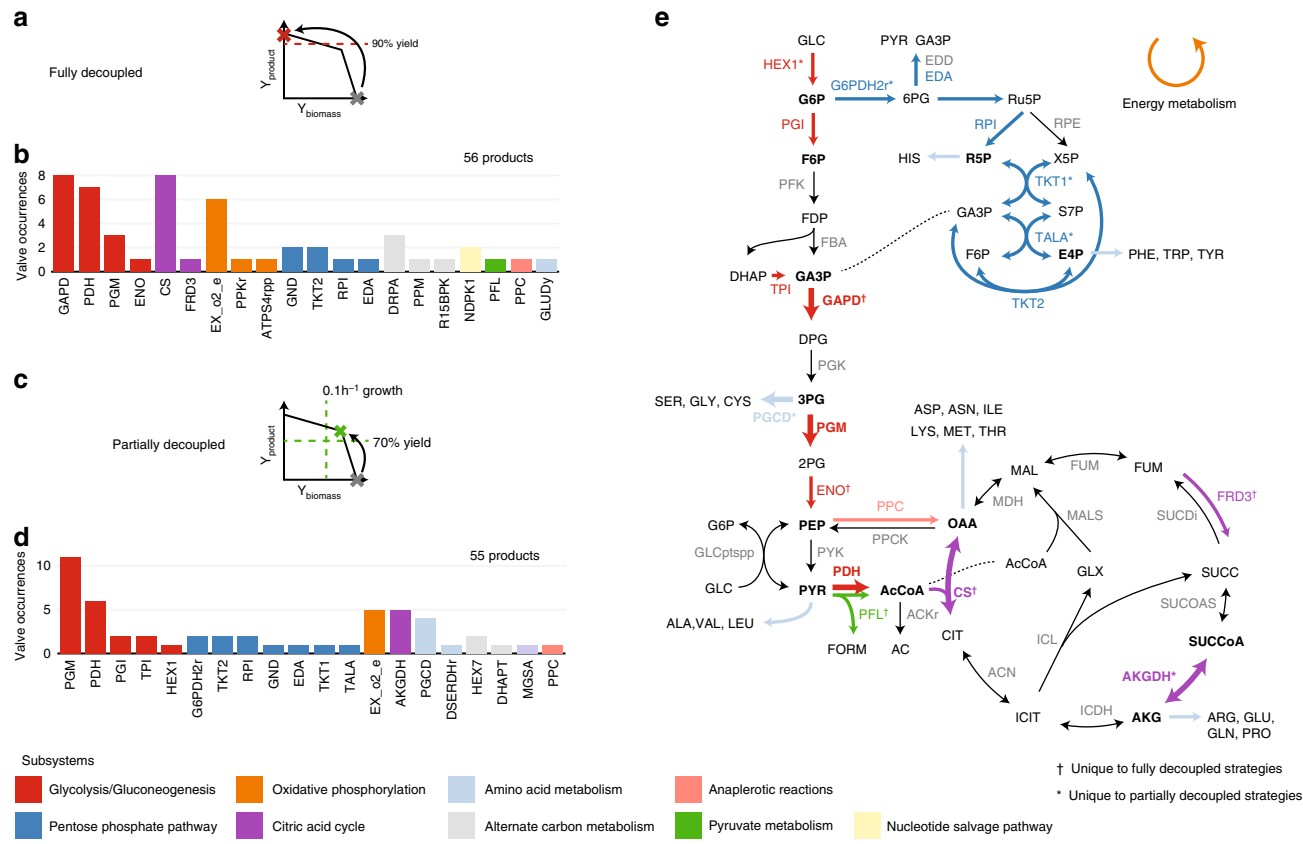

**Fig. 4** Valve occurrences in intervention strategies, categorized by metabolic subsystem. **a** Fully decoupled strategies target over 90% theoretical maximum production without cell growth in the second (production) state. **b** Single valves identified for fully decoupled strategies, single valves can independently redirect flux. **c** Partially decoupled strategies achieve over 70% theoretical yield while maintaining a biomass yield of 0.01 gdw mmol$^{-1}$. **d** Single valves identified for partially decoupled strategies. **e** Central metabolic map highlighting identified valves and associated subsystems. Bolded reactions refer to the top five valves for each decoupling strategy. Bolded metabolites refer to metabolites at the center of the bow-tie architecture of metabolism[34]

in amino acid metabolism is phosphoglycerate dehydrogenase (PGCD), the committing step into serine, cysteine, and glycine metabolism from 3-phosphoglycerate (3PG). It is regulated through feedback inhibition by serine, to maintain appropriate concentrations of these amino acids. PGCD and PGM share a common metabolite (3PG) and were both identified as top valves. By controlling this node, flux can either be directed through PGM toward pyruvate and downstream products, or through PGCD to produce amino acids such as serine and cysteine.

We also identified higher-order strategies which required actuating multiple valves simultaneously. By applying two or three valves, we identified strategies for 64 and 68 products, respectively, compared to 56 products using single valves. This indicates that multiple valves can be required to decouple growth and production for some targets. We also identified clusters of valves which include reactions from a wide range of different subsystems for both fully (Supplementary Fig. 7) and partially (Supplementary Fig. 8) decoupled strategies. These non-intuitive higher-order valves can often be used in conjunction with more intuitive valves to further improve production. Additionally, we have shown that knockouts which were commonly identified amongst all simulations are often found in pyruvate metabolism, to eliminate alternative fermentative byproducts, as well as amino acid metabolism, eliminating alternate routes for flux leakage (Supplementary Fig. 9).

**Genome-scale strategies S. cereivisae.** Finally, we applied MoVE to a genome-scale model of *S. cerevisiae*, a common eukaryotic

production host, to assess the method's effectiveness in a more complex multi-compartment model. Using a similar procedure, we searched for partially decoupled strategies for all 84 metabolites producible from glucose in our model, targeting a minimum biomass yield of 0.001 gdw/mmol (achieving a growth rate of 0.01 h$^{-1}$) and a minimum product yield of 70% of theoretical maximum in the production state. Strategies also targeted over 90% of theoretical maximum growth rate in the growth state. We identified solutions for 61 of the 84 targets using single valves (Supplementary Table 2). Mitochondrial succinate dehydrogenases (SUCD1m, SUCD2_u6m, SUCD3_u6m) were found for over 20 targets, making this an important target in *S. cerevisiae*. In addition, several valves were identified at important branchpoints, similar to *E. coli* (Supplementary Fig. 10).

## Discussion

Here, we have developed a method that can be generically applied to identify metabolic valves to redirect metabolism between phenotypic states, using readily available metabolic models. Using this method, we have shown that decoupling of growth and production phenotypes is possible for a majority of natural chemicals in *E. coli* and *S. cerevisiae*. We have identified strategies to achieve near theoretical maximum product and biomass yield by manipulating three or fewer valves with over 60% of strategies requiring 15 or fewer knockouts (Supplementary Fig. 11), demonstrating the feasibility of two-stage production strategies for diverse targets. Strategies identified by MoVE can be refined through iterative rounds

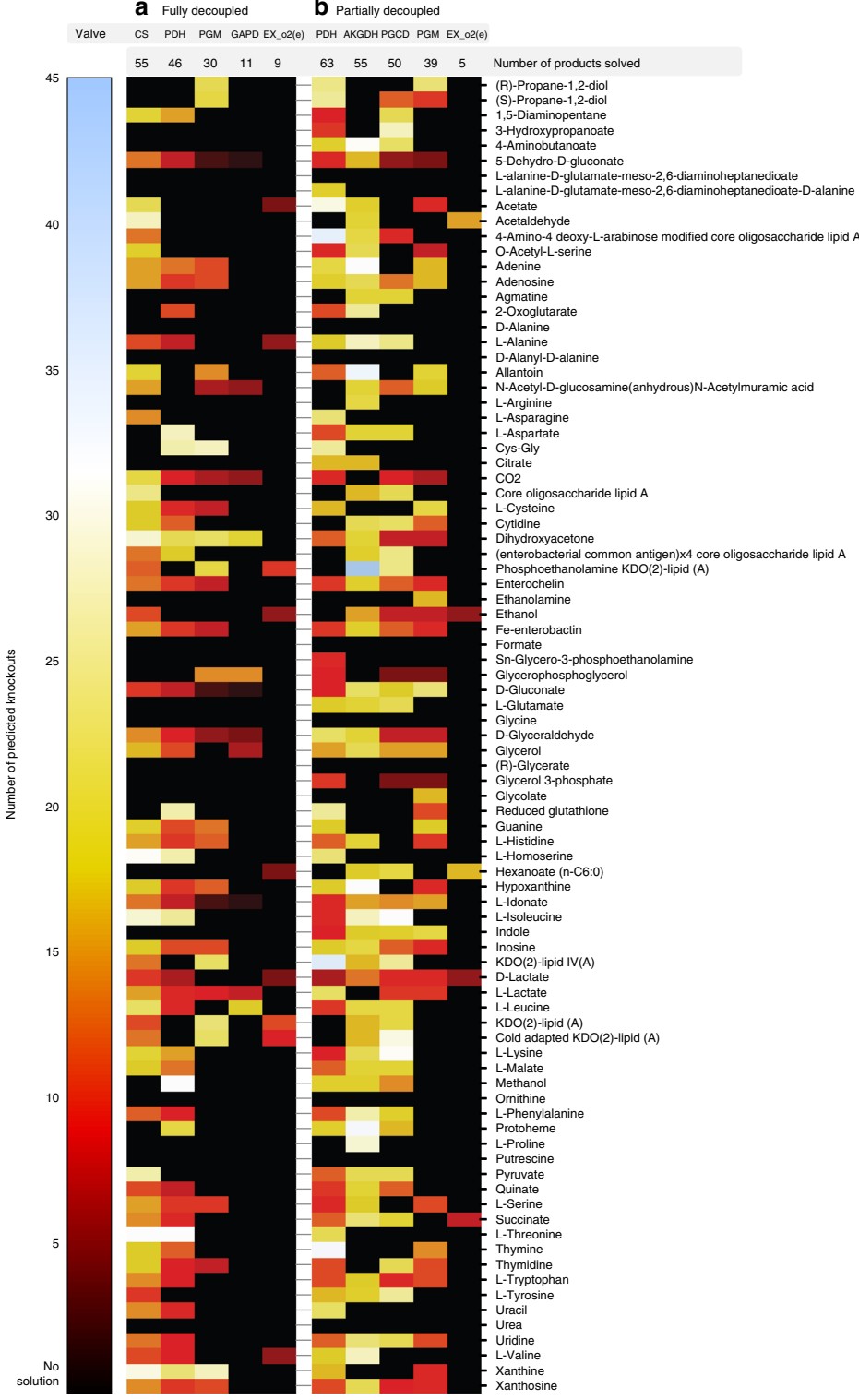

**Fig. 5** Decoupling potential for top five most frequently identified metabolic valves. Columns correspond to strategies employing specified metabolic valves. Rows correspond to specific products. The color scale represents the number of knockouts in the identified strategy for each product and given valve, black indicates no solution was found. **a** Strategies for fully decoupled production. **b** Strategies for partially decoupled production

of experimentation, model refinement and strain design given the inherent biological uncertainty within these models.

We have made this data set fully available to be used as a guide for metabolic engineering endeavors, or to be used as seeds to identify strategies for related compounds. These strategies can be combined with recent methods for strain design prioritization,

based on metrics such as robustness[39], to effectively guide experimental implementation of such dynamic metabolic engineering strategies. Using this comprehensive data set, we identified a high proportion of valves in energy metabolism and near important metabolic bottlenecks, indicating that these bottleneck metabolites are important targets for both natural and synthetic

control. Furthermore, we have identified valves which can be applied to a wide range of products, making them strong candidates for modular platform strains. The location of these valves highlight important architectural traits of metabolism[34], and provide insight into important control points.

We anticipate the application of this algorithm will drive the development of dynamically controlled microbial production hosts and allow the design of more efficient genetic engineering strategies. Furthermore, given the rapidly growing number of curated genome-scale models, and the improving ability for metabolic model generation from -omics data[40–42], MoVE could be applied to elucidate important natural regulatory branchpoints in diverse metabolic systems. This will include more complex microbial and multicellular organisms, such as mammalian systems, where extremely complex regulatory networks exist[43,44].

## Methods

**Stoichiometric metabolic models.** Stoichiometric metabolic models are defined by the reactions present in a given organism, based on genome sequences and experimental validation. Central to this metabolic model is the stoichiometric matrix $\mathbf{N}$, with $m$ rows representing metabolites and $n$ columns representing reactions. Steady state is assumed in constraint-based models, demanding that there is no net accumulation or consumption of internal metabolites:

$$\mathbf{N} \cdot \mathbf{r} = \mathbf{0} \tag{1}$$

where $\mathbf{r}$ is the steady-state flux vector. The network is often further constrained by setting (known) flux bounds for certain reactions, $i$, to define upper bounds on uptake rates (e.g., glucose or oxygen), or fix parameters such as ATP maintenance:

$$\alpha_i \leq r_i \leq \beta_i \tag{2}$$

These flux bounds also include flux directionality constraints for irreversible reactions:

$$r_i \geq 0 \; \forall \, i \in \text{Irr} \tag{3}$$

We assume here that all fluxes in the network are explicitly or implicitly bounded by these constraints (Eqs. 1–3).

**Two-state problem formulation.** The MoVE algorithm aims to find a minimal set of knockouts and a set of dynamically regulated valves to allow switching between two distinct metabolic phenotypes. Here, we have applied this algorithm to achieve efficient switching between two relevant phenotypes for two-stage bioproduction processes: growth and production. This is accomplished by formulating a mixed-integer linear program, which can then be solved using a range of commercial or open-source solvers. Diverse optimization problems, including those relying on stoichiometric models, have been solved in this fashion.

To consider multiple phenotypes in a dynamic context, we require the specification of flux vectors for both of these states and variables describing both static (knockout) and dynamic (valve) interventions. Furthermore, this method must remain scalable to genome-scale models, given these additional variables and constraints. In MoVE, $\mathbf{r}$ denotes the production state (state 2) flux vector and a second flux vector, $\mathbf{f}$, is introduced to represent the growth state (state 1), subject to similar flux bounds: $\gamma_i \leq f_i \leq \delta_i$, and steady-state constraints: $\mathbf{N} \cdot \mathbf{f} = \mathbf{0}$.

We introduce the parameters $\mathbf{T}$ ($t \times n$) and $\mathbf{t}$ ($t \times 1$) to formulate linear inequality constraints for undesired flux vectors in the production state (state 2, e.g., low product yield, Fig. 1c):

$$\mathbf{T} \cdot \mathbf{r} \leq \mathbf{t} \tag{4}$$

For convenience, these inequalities can be formulated to eliminate flux vectors below a minimum yield threshold $\left(Y_{\text{min,state2}}^{\text{P/S}}\right)$, where $r_P$, $r_S$, $r_B$ represent the production or consumption rates of product, substrate and biomass, respectively:

$$\frac{r_P}{r_S} \leq Y_{\text{min,state2}}^{\text{P/S}} \Leftrightarrow r_P - Y_{\text{min,state2}}^{\text{P/S}} \cdot r_S \leq 0 \tag{5}$$

Hence, the matrix $\mathbf{T}$ has a single row of zeros derived from Eq. (5), except for a '+1' in the column for the product reaction, and '$-Y_{\text{min,state2}}^{\text{P/S}}$' in the column for the substrate reaction. The vector $\mathbf{t}$ accordingly contains only one element, '0'.

Similarly, the parameters $\mathbf{D}$ ($d \times n$) and $\mathbf{d}$ ($d \times 1$) impose constraints for the desired flux vectors in the production state (state 2, e.g. high product yield, Fig. 1c):

$$\mathbf{D} \cdot \mathbf{r} \leq \mathbf{d} \tag{6}$$

Again, these constraints are formulated to describe desired product yield:

$$\frac{r_P}{r_S} \geq Y_{\text{min,state2}}^{\text{P/S}} \Leftrightarrow Y_{\text{min,state2}}^{\text{P/S}} \cdot r_S - r_P \leq 0 \tag{7}$$

and biomass yield, in the case of partial decoupling (where some minimum growth rate, $r_B$, is maintained in the production state):

$$\frac{r_B}{r_S} \geq Y_{\text{min,state2}}^{\text{B/S}} \Leftrightarrow Y_{\text{min,state2}}^{\text{B/S}} \cdot r_S - r_B \leq 0 \tag{8}$$

In this case, $\mathbf{D}$ consists of two rows derived from Eqs. 7 and 8: the first contains zeros except for a '$-1$' in the column for $r_P$ and '$Y_{\text{min,state2}}^{\text{P/S}}$' in the column for $r_S$, the second contains non-zero values only for the $r_B$ ($-1$) and again for the substrate uptake rate $r_S \left(Y_{\text{min,state2}}^{\text{B/S}}\right)$. Accordingly, the vector $\mathbf{d}$ is of size 2, and contains '0' elements.

In addition, we also introduce the parameters $\mathbf{G}$ ($g \times n$) and $\mathbf{g}$ ($g \times 1$) to represent desired phenotypes in the growth state (state 1, e.g., high biomass yield, Fig. 1c), which now relies on the growth state flux vector, $\mathbf{f}$:

$$\mathbf{G} \cdot \mathbf{f} \leq \mathbf{g} \tag{9}$$

and is again formulated for yield constraints:

$$\frac{f_B}{f_S} \geq Y_{\text{min,state1}}^{\text{B/S}} \Leftrightarrow Y_{\text{min,state1}}^{\text{B/S}} \cdot f_S - r_B \leq 0 \tag{10}$$

Furthermore, constraints are added to describe desired flux vectors in the production (Eq. (6)) and growth (Eq. (9)) states, to ensure ATP maintenance ($r_{\text{ATPM}} \geq \text{ATPM}_{\min}$ and $f_{\text{ATPM}} \geq \text{ATPM}_{\min}$) and substrate uptake rates ($r_S \leq r_{S,\max}$ and $f_S \leq f_{S,\max}$).

MoVE applies the constraints defining the undesired (Eq. 4) and desired (Eqs. 6 and 9) flux spaces, to identify valves and knockouts.

**Algorithm.** To identify interventions allowing an efficient switch between growth and production states, MoVE applies the concept of minimal cut sets (MCS)[45], a minimal set of knockouts to eliminate undesired functionality. For computation of MCS, the primal problem described above (Eqs. 1–4) is transformed into its dual[18,46–48] and constraints for desired functionality (Eqs. 6 and 9) are applied. Finally, an objective function is used to find solutions requiring a minimal number of interventions. The full formulation of the MoVE optimization problem thus reads:

$$\text{minimize} \quad \sum z_i$$
$$\text{s.t.}$$

$$\begin{pmatrix} \mathbf{N}_{\text{Irr}}^T & \mathbf{I}_{\text{Irr}} & 0 & 0 & \mathbf{T}_{\text{Irr}}^T & 0 & 0 \\ \mathbf{N}_{\text{Rev}}^T & 0 & \mathbf{I}_{\text{Rev}} & -\mathbf{I}_{\text{Rev}} & \mathbf{T}_{\text{Rev}}^T & 0 & 0 \\ 0 & 0 & 0 & 0 & 0 & \mathbf{N} & 0 \\ 0 & 0 & 0 & 0 & 0 & \mathbf{D} & 0 \\ 0 & 0 & 0 & 0 & 0 & \mathbf{N} & 0 \\ 0 & 0 & 0 & 0 & 0 & 0 & \mathbf{G} \end{pmatrix} \begin{pmatrix} \mathbf{u} \\ \mathbf{vp}_{\text{Irr}} \\ \mathbf{vp}_{\text{Rev}} \\ \mathbf{vn}_{\text{Rev}} \\ \mathbf{w} \\ \mathbf{r} \\ \mathbf{f} \end{pmatrix} \begin{matrix} \geq \\ = \\ = \\ \leq \\ = \\ \leq \end{matrix} \begin{pmatrix} 0 \\ 0 \\ 0 \\ \mathbf{d} \\ 0 \\ \mathbf{g} \end{pmatrix}$$

$$\mathbf{t}^T \mathbf{w} \leq -c$$
$$\mathbf{u} \in \mathbb{R}^m$$
$$\mathbf{r}, \mathbf{f} \in \mathbb{R}^n$$
$$\mathbf{w} \in \mathbb{R}^t$$
$$\boldsymbol{\alpha}, \boldsymbol{\beta}, \boldsymbol{\gamma}, \boldsymbol{\delta} \in \mathbb{R}^n$$
$$zp_i, zn_i, y_i \in \{0, 1\}$$
$$\forall \, i \in \text{Rev}: z_i = zp_i + zn_i, z_i \leq 1$$
$$\forall \, i \in \text{Irr}: z_i = zp_i$$
$$\mathbf{vp}_{\text{Irr}}, \mathbf{vp}_{\text{Rev}}, \mathbf{vn}_{\text{Rev}}, \mathbf{w} \geq 0$$
$$c > 0$$
$$r_i \geq (1 - z_i) \cdot \alpha_i$$
$$r_i \leq (1 - z_i) \cdot \beta_i$$
$$y_i \geq (1 - z_i)$$
$$f_i \leq y_i \cdot \delta_i$$
$$f_i \geq y_i \cdot \gamma_i$$
$$\sum_i y_i - (1 - z_i) \leq \text{max\_valves}$$

$$\tag{11}$$

To enable efficient calculation of MCS, new dual variables are introduced, **u**, **w**, **vp_Irr**, **vp_Rev**, **vn_Rev** and the production state variables are further separated into reversible and irreversible components. Following, the stoichiometric matrix **N**, the identity matrix **I**, and the undesired flux matrix **T** are split into two submatrices containing the reversible ($N_{Rev}$, $I_{Rev}$, $T_{Rev}$) and irreversible ($N_{Irr}$, $I_{Irr}$, $T_{Irr}$) reactions (columns).

This makes it possible to use Boolean indicator variables $zp_i = 0 \Leftrightarrow vp_i = 0$, $zp_i = 1 \Leftrightarrow vp_i \neq 0$ for all reactions, and additionally $zn_i = 0 \Leftrightarrow vn_i = 0$, $zn_i = 1 \Leftrightarrow vn_i \neq 0$ for reversible reactions. If the value of an indicator variable is 1, then its associated reaction is in the cut set and can carry no flux as demanded by the constraints for $r_i$.

Although identified MCSs will successfully eliminate undesired functionality (Eq. 4), these MCSs do not guarantee any desired functionality will remain feasible. To do so, the additional constraints for the production (Eq. 6) and growth (Eq. 9) states are applied. In addition, new constraints are added to ensure that metabolic valve reactions are a subset of the reaction knockouts (i.e., flux through valves is ON in the growth state, and OFF in the production state):

$$y_i \geq (1 - z_i) \qquad (12)$$

and to limit the number of possible valves:

$$\sum_i y_i - (1 - z_i) \leq \text{max\_valves} \qquad (13)$$

The Boolean variables $y_i$ thus indicate whether a reaction can carry flux ($y_i = 1$) or not ($y_i = 0$) in the growth state. Up to max_valves reactions of a MCS (as determined by the values of the $z_i$ variables) are allowed to carry flux in the growth state. Hence, the valve reactions are those for which $y_i = 1$ and $z_i = 1$. The reactions for which $y_i = 0$ and $z_i = 1$ are static knockouts and are disabled in both production and growth state, whereas all other reactions are available in both states.

Finally, these variables and constraints are combined to allow the direct identification of optimal combinations of valves and knockouts (with a minimal number of interventions). It is important to note that since any feasible solution will achieve desired functionality (in both states) and eliminate undesired functionality, solving the algorithm to optimality is not absolutely essential.

**Implementation**. Simulations were performed in MATLAB 2010b using the COBRA toolbox[49] and CellNetAnalyzer v2017.4 (CNA)[50]. Mixed-integer linear programs were solved using ILOG CPLEX (IBM, v12.6), via the provided Java virtual machine interface in CNA.

Genome-scale *E. coli* simulations were performed on the SciNet general purpose cluster[51]. The cluster is composed of 3864 nodes using Infiniband interconnect. Each node contains 2x Intel Xeon E5540 processors for a total of 8 cores or 16 threads per node, with 16 GB of RAM. Simulations were performed in parallel for each metabolite and set of parameters. Each simulation was performed on 4 nodes, using 16 threads per node, for 8 h. The MILP was solved in two steps, ramp-up and distributed tree search. In the ramp-up phase, the same problem is solved on each node using different startup parameters for two hours. Following the ramp-up phase, the optimal startup parameters are used to start a distributed search tree for the remaining 6 h. The optimal feasible solution from this process is returned.

Genome-scale *S. cerevisiae* solutions were solved for two hours using 4x Intel Xeon CPU E7-4830 processors, for a total of 32 cores.

**Model and MILP parameters**. The *E. coli* core model[31] was derived from iAF1260[52] and is available from the BiGG database[53] (http://bigg.ucsd.edu/models/e_coli_core). Maximum glucose uptake rate was set at 10 mmol gdw$^{-1}$ h$^{-1}$ and minimum ATP maintenance at 8.39 mmol gdw$^{-1}$ h$^{-1}$.

The *E. coli* genome-scale model iJO1366[33] was used for all genome-scale simulations and available from the BiGG database (http://bigg.ucsd.edu/models/iJO1366). Maximum glucose uptake rate was set at 10 mmol g dw$^{-1}$ h$^{-1}$ and minimum ATP maintenance at 3.15 mmol gdw$^{-1}$ h$^{-1}$. Target reactions were chosen from the total set of export reactions by eliminating non-organic molecules and those which were not producible from glucose based on a flux variability analysis. A set of exchange reactions, excluding the target reaction, common fermentation products, and non-organic molecules, was removed from the model to improve computational feasibility; this list is provided in Supplemental Information.

Strategies for the *E. coli* core model were solved to optimality using an upper bound of three valves in negligible computational time. Strategies for the genome-scale model were solved using the distributed MILP search method with an equality constraint on the number of valves. Searches explicitly specified one, two, or three valves. To identify the optimal valve for each metabolite, all reactions were allowed to be used as valves for fully coupled strategies, and only non-essential valves for partially coupled strategies. To identify valves which could be applied to many products, we ran independent optimizations with each valve explicitly specified (no other valve was allowed to be identified as a valve).

Strategies for the *S. cerevisiae* genome-scale model were identified using iMM904[54] which is available from the BiGG database (http://bigg.ucsd.edu/models/iMM904). Maximum glucose uptake rate was set at 10 mmol gdw$^{-1}$ h$^{-1}$ and minimum ATP maintenance at 1 mmol gdw$^{-1}$ h$^{-1}$. A set of exchange reactions, excluding the target reaction and reactions required for wild-type growth, was removed from the model to improve computational feasibility; this list is provided in Supplemental Information.

**Reaction connectivity**. Reaction connectivity was determined in Python 3.5 using the COBRApy package[55]. Connectivity for each reaction was determined as the sum of the connectivities of all metabolites involved in that reaction. The connectivity of each metabolite is determined as the number of reactions in which it partakes.

**Clustering of higher-order valve strategies**. Intervention strategies were determined using one, two, or three metabolic valves. Higher-order valves which improved the objective (e.g., a solution was only found with a larger number of valves, or the required number of knockouts was decreased) were used to generate an adjacency matrix for clustering, using the co-occurence frequency as a similarity metric. The resulting sparsely connected matrix was clustered using an iterative spectral clustering approach. Several iterations of spectral clustering were performed using the scikit-learn package (v0.19.0). Following, a new similarity matrix was generated based on the mutual information available between clustering solutions to identify the most represented solution, this clustering result is returned. The numbers of clusters was chosen to ensure clusters contained at least two members.

**Code availability**. The MoVE algorithm is available at https://github.com/lmse/move and as Supplementary Software 1.

## Data availability

The authors declare that the data supporting the findings of this study are available within the paper and are included as Supplementary Information. A summary of simulation results is provided as Supplementary Data 1, raw results (knockouts and valves) of all simulations are available as Supplementary Data 2, Python-readable data is provided as Supplementary Data 3, and the list of knocked out reactions in simulations to improve computational efficiency is provided as Supplementary Data 4.

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

## Acknowledgements

We acknowledge Daniel Tomchynsyn and Dean Robson for support in deploying this algorithm, as well as Compute Canada and SciNet for providing compute resources. We would also like to thank Elad Noor for insightful discussions. This work was supported by the Natural Sciences and Engineering Research Council (NSERC), the Industrial Biocatalysis Network, BioFuelNet Canada, the Ontario Ministry of Research and Innovation, the Alexander von Humboldt foundation, the German Federal Ministry of Education and Research (FKZ: 031L104B) and the European Research Council (ERC Consolidator Grant 721176).

## Author contributions

N.V. and R.M. conceived the study. A.vK. and N.V. implemented the algorithm. N.V. performed calculations and analyzed the results. N.V., A.vK., S.K., and R.M. discussed the results and wrote the manuscript.

## Additional information

**Competing interests:** The authors declare no competing interests.

