## [Peer Review File · Nature Communications]

Reviewers' comments:

Reviewer #1 (Remarks to the Author):

Summary: Venayak et al developed the metabolic valve enumerator (MoVE) tool to identify key reactions (valves) whose interruption enables metabolic transition from the maximum growth (or biomass synthesis) phase to the no-growth maximum product synthesis phase (full decoupling) or to the sub maximum product synthesis phase with some allowable growth (partially decoupling).

General comments: Growth decoupling strategy for enhanced product formation during the no-growth phase has been widely applied in industrial processes from production of bulk chemicals to high-value chemicals. Therefore, this paper tackles a very relevant and important problem to find the best metabolic valves to decouple these two metabolic processes to achieve high product yields and rates. Analysis of common valves is very intriguing. The paper formulates a sound optimization problem and algorithms to implement MoVE.

Major comment: Transition from the maximum growth phase to maximum product synthesis phase as formulated in the study might be impractical to implement, especially when aiming to make heterologous products. To make products in the no-growth phase, enzymes must be readily available. This synthesis can only be achieved during the growth phase because protein synthesis is tightly coupled with growth. Therefore, desirable products must be synthesized during the growth phase to a certain extent. As a result, the constraint, as outlined in the paper, that the design seeks to achieve the maximum growth during the growth phase only (without product formation) will not likely help maximize the product synthesis during the no-growth phase. To circumvent this, it is more reasonable for the authors to formulate the problem as a transition from the weak growth-product coupling phase to the no-growth, maximum product synthesis phase.

Minor comments:

1. Abstract should not include references
2. "Natural" products are typically referred to secondary metabolites. To avoid confusion, the authors might use "endogenous" products for "natural" products and "heterologous" products for "non-natural" products.

Reviewer #2 (Remarks to the Author):

The authors report an MILP algorithm to predict key metabolic nodes useful for engineering dynamic metabolic states specific to a given network/product constraints. Importantly, to the knowledge of this reviewer, the reported approach is the first systematic optimization method for dynamic metabolic engineering which can not only be applied to fully decoupled states but also partially decoupled states.

1. While the approach and algorithm may be useful and novel, the authors present little validation of the predictions made. While complete experimental validation is unrealistic, some level of validation, even if this is providing understanding to recently published results?

2. Approaches such as Optknock used to optimize growth associated (GA) production strains, rarely work a priori. Almost uniformly strains need to be experimentally optimized once deletions are made to "rewire" expression and activity levels in the metabolic network to drive to the optimal solution. Fortunately, GA processes can rely on the power of directed evolution including adaptive laboratory evolution to reach these more optimal production levels. A limitation of dynamic control is the lack of the ability to select for better producers and other non-predictable changes to the metabolic network, and so the predictions made by MoVE alone are unlikely to give the optimal results. The authors should further discuss how MoVE can be used in a practical way

and how they would propose to address this issue in engineering production strains.

3. As the authors are correct that CS is a very "intuitive valve", (as are many of the other valves that are listed), a more detailed discussion of how the Move Approach and its results give insight above and beyond the "intuitive choice" of valves is important to clearly set the need and potential advantage for such an algorithm. The statement is true also for the global role of the PEP/pyruvate node. Is a key understanding of the work is that the major branch/control points in metabolism that one would intuitively choose are "confirmed" by Move or are there new key insights that the approach is bringing to light. If so what are these insights.

4. Higher order valve strategies. As individual valves may be intuitive, the higher order strategies may be of more importance in optimization and less obvious. It appears to this reviewer from the methodology used that the higher order valve strategies represent only one set of options but not a global optimum/maximum, please clarify. If an optimum is not identified for combinatorial approaches, what could be missed, what are the benefits of this approach and route and what alternatives can be investigated computationally? This perhaps one of the potentially most innovative and usefule aspects of Move which needs further discussion, clarification and validation.

Minor Comments

1. For the genes names, please use the full gene name in the text before the abbreviation such as citrate synthase before CS.
2. Are there reasons for the choice of constrains (growth rate and yield) for the the partially decoupled optimizations, please discuss.

Reviewer 1

Major concerns

Transition from the maximum growth phase to maximum product synthesis phase as formulated in the study might be impractical to implement, especially when aiming to make heterologous products.

To make products in the no-growth phase, enzymes must be readily available. This synthesis can only be achieved during the growth phase because protein synthesis is tightly coupled with growth.

Therefore, desirable products must be synthesized during the growth phase to a certain extent. As a result, the constraint, as outlined in the paper, that the design seeks to achieve the maximum growth during the growth phase only (without product formation) will not likely help maximize the product synthesis during the no-growth phase. To circumvent this, it is more reasonable for the authors to formulate the problem as a transition from the weak growth-product coupling phase to the no-growth, maximum product synthesis phase.

We would like to thank the reviewer for providing this feedback and their time. While it is true that the enzymes for the product synthesis needs to be present in order to allow the transition to the maximum product phase, our algorithm does not force the cells to operate in maximum growth scenario. Rather, our algorithm allows for the maximum growth rate (or 90% of the maximum) to be reached in the growth phase. We want to point out that this does not preclude the expression of genes corresponding to the product synthesis pathway even though it is likely that the substrate will be effectively channeled to the growth pathways in the first phase. Hence, during the implementation, one can choose to express the production pathways during the growth stage in order to allow the enzymes to be present. Furthermore, previous studies have shown that even the best weak growth coupled designs suffer from more than a 10% drop in the growth rate that could affect the growth of the cells in the growth phase. Finally, the production of metabolic pathways towards the desired product can be continued during the production stage by using a partially decoupled strategy, where some cell growth (and thus the ability to produce amino acids) is maintained. We have incorporated in the manuscript a discussion on this topic of expressing enzymes for the product pathway in the growth phase (Lines 80-83).

Minor comments

1. Abstract should not include references **Abstract has been modified**
2. “Natural” products are typically referred to secondary metabolites. To avoid confusion, the authors might use “endogenous” products for “natural” products and “heterologous” products for “non-natural” products.

We appreciate the suggestion for alternative nomenclature. Although the nomenclature is not consistent throughout the literature, we tend to follow the nomenclature provided in the following reference. Hence, we have replaced “natural products” with “natural chemicals”:

J. W. Lee *et al.*, Systems metabolic engineering of microorganisms for natural and non-natural chemicals. *Nat. Chem. Biol.* **8**, 536–46 (2012).

Reviewer #2

Major concerns

1. While the approach and algorithm may be useful and novel, the authors present little validation of the predictions made. While complete experimental validation is unrealistic, some level of validation, even if this is providing understanding to recently published results? We would like to thank the reviewer for their valuable and highly constructive feedback. These points are very useful for us to revise our paper to strengthen it.

The reviewer is correct to point out that complete experimental validation would support the conclusions gained from the algorithm simulations, but is infeasible on the scale of this study. We would like to point out that we have included a very relevant study of a strategy which can be identified by MoVE and has successfully been implemented. We have expanded on this study in the discussion (lines 58-63). We believe this motivates the applicability of MoVE in a broader context. We also note that there were several recent presentations at conferences that further support our predictions even though they are not yet published.

2. Approaches such as OptKnock used to optimize growth associated (GA) production strains, rarely work a priori. Almost uniformly strains need to be experimentally optimized once deletions are made to “rewire” expression and activity levels in the metabolic network to drive to the optimal solution. Fortunately, GA processes can rely on the power of directed evolution including adaptive laboratory evolution to reach these more optimal production levels. A limitation of dynamic control is the lack of the ability to select for better producers and other non-predictable changes to the metabolic network, and so the predictions made by MoVE alone are unlikely to give the optimal results. The authors should further discuss how MoVE can be used in a practical way and how they would propose to address this issue in engineering production strains.

This reviewer clearly points out one of the key underlying assumptions of the OptKnock algorithm (adaptive evolution). This algorithm (as other modern algorithms) consider a different problem than OptKnock: where OptKnock relies on maximizing the growth-coupled yield, the minimal cutset (MCS) algorithms and MoVE algorithm rely on achieving a *minimum* production target. Thus, this target need not rely on adaptive evolution directly. However, in the case of partial decoupling, adaptive evolution can be applied in exactly the same fashion as OptKnock strategies since these strategies allow for cell growth. We present motivation for both strategies for future implementation. In addition, adaptive evolution can be applied to improve the growth rate in the growth stage as described the main manuscript as Figure 2 (previously SI Figure 1). We now refer to this implementation in the main text (L51-52).

3. As the authors are correct that CS is a very “intuitive valve”, (as are many of the other valves that are listed), a more detailed discussion of how the Move Approach and its results give insight above and beyond the “intuitive choice” of valves is important to clearly set the need and potential advantage for such an algorithm. The statement is true also for the global role of the PEP/pyruvate node. Is a key understanding of the work is that the major branch/control points in metabolism that one would intuitively choose are “confirmed” by Move or are there new key insights that the approach is bringing to light. If so what are these insights.

The identity and topology of the metabolic valves for a range of diverse products validate the intuition of many of such *intuitive valves*. Although these may have been logical choices, it is not so obvious when considering maintaining the balanced stoichiometry of reactions (which, in our view, is one of the most important applications of metabolic models). In addition, one of the key insights from our study is the location of these valves close to the 12 key precursor nodes in the bow tie structure of the metabolic network. Note that these 12 nodes are not necessarily the highly connected nodes (SI Figure 3) but are in a strategic location in the network that allows control over the biomass production pathways. Furthermore, although we focused on metabolic valves since this is the unique portion of this study, we should point out that each of these valves has an associated set of knockouts which is surely non-intuitive and required to enable the identified valves.

We did also identify several valves in central metabolism and additionally many higher-order strategies (which are surely much more difficult to intuit) which provide more insight into metabolism. These results are presented in the supplementary information (Figure 8 and Figure 9). Furthermore, one example of a product that can be made using an intuitive valve and a non-intuitive valve is shown in the table below (pulled from SI tables). However, we would also like to point out that it is difficult for us to identify any key insights or design principles underlying these non-intuitive valves even though we have tried hard to rationalize these valves. We have added a discussion along these lines in our paper in order to address this comment (Lines 145-147).

EX_ser-L(e)	Knockouts	ETHAAL	GLYCL	LDH_D	MGSA	PFL	POX	PPC	THD2pp
	Valves	PDH	RPI						

4. Higher order valve strategies. As individual valves may be intuitive, the higher order strategies may be of more importance in optimization and less obvious. It appears to this reviewer from the methodology used that the higher order valve strategies represent only one set of options but not a global optimum/maximum, please clarify. If an optimum is not identified for combinatorial approaches, what could be missed, what are the benefits of this approach and route and what alternatives can be investigated computationally? This perhaps one of the potentially most innovative and usefule aspects of Move which needs further discussion, clarification and validation.

Following from above, we would like to indicate that the solution we discovered with MoVE are not global optimal solutions. However, it is important to note that the objective function here is the number of valves, with a maximum constraint on number of valves (1, 2, or 3, in this study). In other words, even the “non-optimal” solutions we find satisfy the stringent product requirements. The reason the solution might be non-optimal is that there may be exist a solution that has fewer valves and knock-outs. Given the scale of this study (where we solved for all products in *E. coli*), we found the best feasible solution within a fixed computational time (32 cores, for 8 hours). With a single product in mind, further computational time could be expended to find global optimal solutions. We should add that although the solver was not able to prove optimality, in most cases the objective function reached its minimal value within only a fraction of the total expended computation time, so these solutions are good candidates with minimal interventions required.

Furthermore, given a single product one could choose to enumerate ALL solutions to the problem, providing further insight into metabolism around the given product of interest. We have

incorporated these discussion points into the revised manuscript in order to address these important points raised by the reviewer.

In addition, most of the solutions (> 60%) we have found require less than 15 deletions and valves (Lines 164-165). We feel that this will allow majority of our solutions to be experimentally implementable given the recent advances in experimental tools for synthetic biology.

Minor Comments

1. For the genes names, please use the full gene name in the text before the abbreviation such as citrate synthase before CS.

This is present (Ln 84).

2. Are there reasons for the choice of constraints (growth rate and yield) for the partially decoupled optimizations, please discuss.

These were set empirically based on some simulations of what thresholds could be reasonably reached given the computational and time constraints available. We would like to add that 90% growth and yield targets are very stringent, and solutions could only be found to reach this target given the scale of the computational resources used in this study.

REVIEWERS' COMMENTS:

Reviewer #1 (Remarks to the Author):

The authors have addressed my comments. This paper contributes a useful prediction tool to metabolic engineering/synthetic biology communities for strain engineering for production of molecules.

Reviewer #2 (Remarks to the Author):

The authors have made changes to clarify the methodology and add reference to experimental work that supports the study outcomes. The authors have not really addressed a few major issues with the manuscript.

First the issue on implementation of the method. While the authors have commented that adaptive evolution may be used, this feedback was meant as an example only. If a scientist or group was to use the MoVE method to identify valves and implement them and only see marginal results what would there next "move/step" (no pun intended) be? In the case of algorithms like Optknock directed evolution is the next step after implementing the predicted genomic modifications, assuming that further engineering is required. What is the equivalent in the case of MoVE (since you have decoupled growth from production) when after implementing the predicted valves, the performance is still not optimal. It is important to put this approach in the context of the entire engineering program and how it fits in.

Secondly, the authors have not added strong enough language regarding the fact that this algorithm does not generate a global optimum and we expect that significant further engineering (how to be explained) to be done after using the computational approach. Again the concern is that the manuscript does not give context of how this method fits into a more comprehensive engineering program

REVIEWERS' COMMENTS:

Reviewer #1 (Remarks to the Author):

The authors have addressed my comments. This paper contributes a useful prediction tool to metabolic engineering/synthetic biology communities for strain engineering for production of molecules.

Reviewer #2 (Remarks to the Author):

The authors have made changes to clarify the methodology and add reference to experimental work that supports the study outcomes. The authors have not really addressed a few major issues with the manuscript.

First the issue on implementation of the method. While the authors have commented that adaptive evolution may be used, this feedback was meant as an example only. If a scientist or group was to use the MoVE method to identify valves and implement them and only see marginal results what would there next “move/step” (no pun intended) be? In the case of algorithms like OptKnock directed evolution is the next step after implementing the predicted genomic modifications, assuming that further engineering is required. What is the equivalent in the case of MoVE (since you have decoupled growth from production) when after implementing the predicted valves, the performance is still not optimal. It is important to put this approach in the context of the entire engineering program and how it fits in.

This reviewer brings up a general concern for strain design algorithms, and modeling in almost all parts of science: drawing hypotheses based on an inaccurate or incomplete model of the system. In this case, there are two options: improve the model, or understand their limitations to move forward despite them. Although OptKnock proposes adaptive evolution to reach the predicted phenotype, by no means would this work in all cases. Because of our limited understanding of biology and metabolism, metabolic engineering typically proceeds cyclically through design, build, test and learn stages. As such, algorithms such as MoVE (or OptKnock) simply help design strains, but these designs should be refined as more information is gathered. Ideally, this information should be used to improve the metabolic models from which the designs were generated. Alternatively, MoVE algorithm could be used to identify valves that allow growth in the first stage and growth coupled production (with a high product threshold yield, e.g., 75% of the maximum theoretical yield) in the second stage. One could then adaptively evolve the mutant in the second stage and then turn the genes on in the growth stage. We have added a sentence in the discussion on this topic (L 163).

Secondly, the authors have not added strong enough language regarding the fact that this algorithm does not generate a global optimum and we expect that significant further engineering (how to be explained) to be done after using the computational approach. Again the concern is that the manuscript does not give context of how this method fits into a more comprehensive engineering program.

We would like to clarify that the algorithm itself can provide a global optimum (in the context of mathematics and linear programming); however, due to the scale of our study we did not have the computational resources to solve all 500+ simulations to their global optimality. We would also like to

add that although we did not run the algorithm sufficiently to prove a global optimal solution, we noted that the objectives function (minimizing the number of knockouts) did not improve after approximately 20% of our total simulation time (indicating that many of these solutions may have been close to optimal). In addition, we don't believe that the lack of global optimum is a limitation because if a particular target and the host is chosen, one can use the algorithm to identify a global optimum. Furthermore, it must be said that the identifying a global optimum would be an issue for almost all strain design algorithms including the ones that rely on growth coupled production. Nevertheless, we have included in the SI a note (L 257) to this point.

We would also like to elaborate on how this algorithm would fit into a typical metabolic engineering program, and comment on an 'optimal' solution in the context of 'biological correctness', as opposed to mathematical optimality. We provide an example metabolic engineering program for the production of chemical X, to highlight where MoVE could be used:

Pathway identification

- 1) Identify pathway to compound X (using a pathway predictor, for example)
- 2) Identify suitable enzyme families for this pathway
- 3) Screen/engineer enzyme variants for activity and/or specificity

Systems metabolic engineering

- 4) Refine our most current metabolic model to include the reactions added to our production strain
- 5) **Run MoVE to identify gene knockout strategies to optimize flux to our product of interest**
- 6) Implement above strategy, and phenotype strains to understand byproduct profile and metabolic flux (using ¹³C-Metabolic flux analysis, for example)
- 7) Use phenotyping results to improve the model and re-run the algorithm with this improved model OR understand the limitation in the algorithmic prediction that prevented the optimal phenotype being realized (rogue flux due to promiscuous enzyme activity which was not included in the model, for example) and proceed accordingly (remove these enzymes *in silico* or *in vivo*).

We did not elaborate on such a workflow because we believe this is tackled in numerous review articles – including some of the limitations of metabolic modeling noted above.